# An Analysis of Transferability in Network Intrusion Detection using Distributed Deep Learning

**Shreya Ghosh, Abu Shafin Mohammad Mahdee Jameel & Aly El Gamal**
Department of Electrical and Computer Engineering
Purdue University
West Lafayette, IN 47907, USA
`{ghosh64,amahdeej,elgamala}@purdue.edu`

## Abstract

In this paper, we utilize a distributed deep learning framework to investigate transferability of network intrusion detection between federated nodes. Transferable learning makes intrusion detection systems more robust to rare attacks and enables them to adapt to real life scenarios. We analyze symmetric and asymmetric transferability relationships. We propose and investigate the impact of feature pre-processing to improve transferability. The code for this work is available at https://github.com/ghosh64/fedlearn.

## 1 Introduction

Network intrusions are one of the most common threats to Internet of Things (IoT), and Intrusion Detection Systems (IDS) are a vital part of these systems. Devices over a network can be exposed to different types of attacks, and the datasets are generally highly imbalanced with a lot more benign data compared to attack data. Centralized Deep learning based IDS implementations have become popular due to their superior performance. However, a federated learning based method—where training happens on edge nodes and then aggregated—provides better privacy and needs orders of magnitude lower data bandwidth. In such a scenario, it becomes important to be able to propagate training of one kind of attack data from one node to another. In this paper, we present a preliminary framework to analyse and improve such transferability, and propose pre-processing steps to improve the transferability and privacy.

## 2 Existing Work

Traditional network intrusion systems utilized conventional anomaly detection or classifier algorithms (Lansky et al. (2021)). Deep learning based approaches have gained popularity due to their superior performance (Javaid et al. (2016); Shone et al. (2018)) and ability to perform well in the presence of imbalanced data sets (Fu et al. (2022)). Current developments have also highlighted the possibility to develop low resource deep learning based intrusion detection systems that overcome one of the major shortcomings of a deep learning based approach compared to traditional approaches (Rizvi et al. (2023)). Due to the highly imbalanced nature of the intrusion data, there have been efforts to utilize transferability of learning to train for novel attacks (Catillo et al. (2022)).

In Rahman et al. (2020), it was shown that federated learning approaches to intrusion detection in IoT outperforms self learning, while reaching comparable accuracy compared to a centralized system. This distributed approach preserves privacy compared to centralized deep learning systems. Similar to centralized systems, transferability analysis of federated intrusion detection systems have also received recent attention as it can help improve the adaptability of these systems, and improve their robustness in diverse real life applications (Zhang et al. (2022); Xue et al. (2022)).

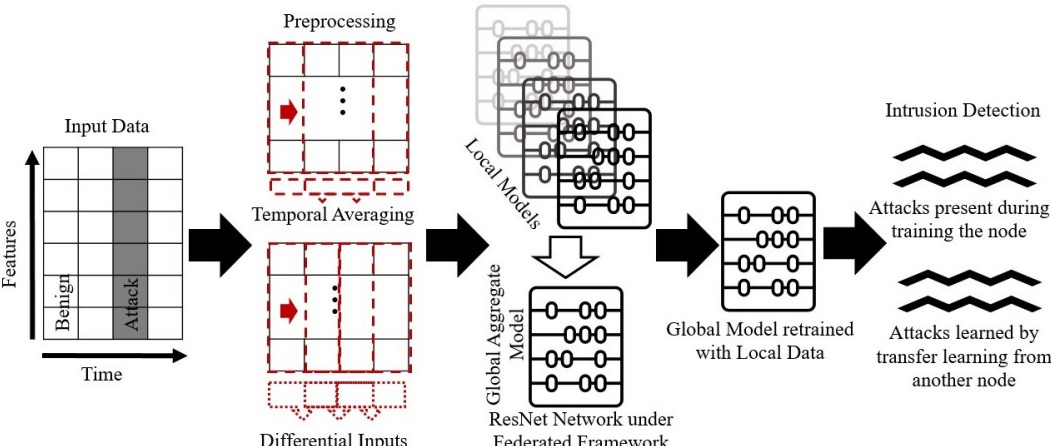

Figure 1: Architecture of Proposed Intrusion Detection System

Table 1: Transferability Study (Percentage of correctly classified attack data)

| (Node, Training Attack, Testing Attack) | (1,1,4) | (5,4,1) | (5,4,3) |
|---|---|---|---|
| No Preprocessing | 71.5 | 72 | 42.5 |
| Temporal Averaging | 93 | 73.6 | 64.5 |
| Differential Inputs | 70.2 | 75.6 | 65 |

## 3 METHODOLOGY

In Fig. 1 we present an overview of the learning framework. The dataset we use has flow, architecture and protocol features from network traffic data, from benign and different types of attack classes. We organize the data to different edge nodes, with different nodes being subject to different kinds of attacks. For this work, we utilize a ResNet network as the deep learning network. Once all the node models are trained, they are sent to the central server and aggregated. Then a global model is sent back to each of the nodes, and this continues for several learning rounds. At the end, each of the nodes do another round of re-training with local data, in order to best adapt to the diverse local environments. This model is then used to detect intrusions at different nodes. To further improve the transferability, we propose two separate pre-processing steps, temporal averaging and differential inputs. These steps have the added impact of improving the privacy of the training data.

### 3.0.1 FEDERATED SETUP

To emulate a real scenario, we consider a setup with five edge devices that are trained with independent and uncorrelated local data. We used FedAvg aggregation algorithm (McMahan et al. (2016)):

$$\forall k, w_{t+1}^k \leftarrow w_t - \eta g_k; w_{t+1} \leftarrow \sum_{k=1}^{K} \frac{n_k}{n} w_{t+1}^k$$

Here, $w_t$ are the model weights after communication round $t$, and we have $K$ total nodes. Appendices A.1 and A.2 have more details.

## 4 RESULTS AND DISCUSSION

In Table 1, we present some selected results from the transferability study. Here, each node is trained on only one kind of attack data (please see Appendix A.3 for a description of the data and nodes), and then after the federated training and local re-training, we test a node on an attack type that it did not see before. For example in Table 1, (1,1,4) means that node 1 was trained on Attack 1 (Denial of Service(DoS)) and then tested with Attack 4 (Probing) and then we present the accuracy under different conditions in the corresponding column. As node 5 was trained with Attack 4, we can see the transferability of that training to node 1 in the federated framework. We can see from the results that in addition to improving the privacy of the data, the proposed pre-processing steps also improve the transferability. In the future, we plan to utilize these observations to develop a more robust IDS.

URM STATEMENT

The authors acknowledge that at least one key author of this work meets the URM criteria of ICLR 2023 Tiny Papers Track.

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

## A  APPENDIX

### A.1  FEDERATED LEARNING

In federated learning, there are several edge devices that train models instead of a single centralized model. A global parameter server chooses edge devices for learning, and sends the global model to the edge devices. The edge devices train the global model on local data, thereby preserving data privacy. The models return trained weights to the global parameter server that aggregates weights to update the global model for that communication round. The global model is then updated for each edge device before the next communication round.

Table 2: Devices and Training Data

| Device | Attack |
|--------|--------|
| Device 1 | Denial of Service(DoS) (Attack 1) |
| Device 2 | Denial of Service(DoS) (Attack 1) |
| Device 3 | User to Root(U2R) (Attack 2) |
| Device 4 | Remote to Local(R2L) (Attack 3) |
| Device 5 | Probing (Attack 4) |

## A.2 NETWORK PARAMETERS

Each device model consists of a model architecture with a pretrained resnet-18 backbone that has been fine-tuned to NSLKDD dataset using a lower learning rate(optimizer-Adam, learning rates: 1e-5 for the resnet backbone, 1e-4 for other layers). Each local model has been trained for 20 epochs per local training and the federated setup was trained for 20 communication rounds.

## A.3 FEDERATED DATASET

We use the NSLKDD dataset for our transferability study. The NSLKDD datset consists of one benign class and 4 attack classes. In our setup, there are 5 devices. Each device is trained with a different kind of attack. Since we have 5 devices and 4 kinds of attacks, devices 1 and 2 are trained with the same attack. Benign data is split equally between all 5 devices. Table 2 illustrates attack classes each device is trained with.

