# OpenReview forum: "An Analysis of Transferability in Network Intrusion Detection using Distributed Deep Learning"
_ICLR.cc/2023/TinyPapers — Submitted to Tiny Papers @ ICLR 2023_

### Official Review · Reviewer_wdzf · 2023-03-23

**Confidence:** 3

**Summary Of Contributions:**

The authors propose utilizing a distributed deep learning framework to investigate transferability of network intrusion detection between federated nodes. Their preliminary studies investigate the impact of feature pre-processing to improve transferability.

**Rating:**

High Potential (HP): a submission which meets the reviewing criteria and has potential to make an impact on the field

**Strengths And Weaknesses:**

Strengths:
- Interesting idea of potentially high impact.

Weaknesses:
- No real description regarding the preprocessing steps is provided.
- Are there any reasons why the temporal averaging seems to be “best” in some sense? Does the differential inputs step really help with attack 1? Remains slightly unclear.


**Suggested Changes:**

- Make sure citations read correctly, i.e., show up in brackets as opposed to regular inline text.
- Provide more information about the preprocessing steps and how they help specifically (maybe in the appendices?).
- Eventually the authors are encouraged to compare against standard methods in the literature to see how their method compares with others.
- Would be nice to have a sense for how much added computation the two suggested preprocessing steps introduce.
- Furthermore, does using a distributed DL system introduce new problems compared to other more conventional techniques?

---

### Comment · Area_Chair_Vf8V · 2023-06-05
**Ready to archive**

This work meets the threshold for archival, contains the URM statement, and is deanonymized.

---

### Meta-Review · Area_Chair_Vf8V · 2023-04-06

**Recommendation:** Invite to present
**Confidence:** 3

**Metareview:**

Thank you for your submission. As the reviewers have noted, this work studies an interesting problem with potentially high impact results. While the reviewers have noted that it does need some additional details and discussion for increased clarity, the suggested changes seem minor to incorporate. Otherwise, the paper seems to be clear, correct, and reproducible.

**Summary:**

This work investigates the transferability of network intrusion detection models in a distributed setting.

**Reason For Not Giving A Higher Recommendation:**

- Requires additional details on preprocessing technique
- Clarity could be increased with additional discussion on the intuition of the results obtained

**Reason For Not Giving A Lower Recommendation:**

- High impact idea
- Well written
- Code is provided

---

### Decision · Program_Chairs · 2023-04-08

Invite to present